Acclimation and degradation characteristic of the microbial system in corn straw

Yu Jiang 1
Wang Chun 1
Qi Yuena 1
Lian Jie 1
Yu Zhenhua 2
Fan Guoquan 3
Sun Jianming 1
Liu Xitao 1
Liu Xiaofei liuxiaofei72@163.com 1
1 Harbin University of Commerce , Harbin , China
2 Northeast Institute of Geography and Agroecology, Chinese Academy of Sciences , Harbin , China
3 Crops Institute, Heilongjiang Academy of Agricultural Sciences , Harbin , China
Brygadyrenko Viktor
Electronic publication date: 2025 Dec 16
Publication date: 2025
Volume: 13
Electronic Location ID: e20386
Received 2025 May 16; Accepted 2025 Oct 22
Copyright: ©2025 Yu et al.
Copyright year: 2025
Copyright holder: Yu et al.
License: This is an open access article distributed under the terms of the Creative Commons Attribution License, which permits unrestricted use, distribution, reproduction and adaptation in any medium and for any purpose provided that it is properly attributed. For attribution, the original author(s), title, publication source (PeerJ) and either DOI or URL of the article must be cited.
License URL: https://creativecommons.org/licenses/by/4.0/

Keywords: Lignocellulose, Microbial system, Corn straw, Functional analysis, Degradation mechanism

Funding: Heilongjiang Provincial Institute Research Expenses CZKYF2024-1-A005 Heilongjiang Province Postdoctoral Funding Project LBH-Z23209 This research was funded by Heilongjiang Provincial Institute Research Expenses project (CZKYF2024-1-A005) and Heilongjiang Province Postdoctoral Funding Project(LBH-Z23209). The funders had no role in study design, data collection and analysis, decision to publish, or preparation of the manuscript.

==============================
Background

Lignocellulose represents a significant biological resource in corn straw, yet its efficient degradation presents challenges.

Purpose

This study aimed to enhance the degradation rate of lignocellulose to expand the utility of corn straw.

Methods

A microbial system capable of degrading corn straw was acclimatized at an initial temperature of 30 °C using a static temperature limiting cultivation method.

Results

During the biodegradation process via fermentation, the concentration of organic acids peaked on the third day before gradually declining. Concurrently, the concentration of soluble sugars reached a maximum of 8.9 µg mL−1 on the third day, while the total sugar content swiftly reduced from 4 mg mL−1 to 1.6 mg mL−1 within the initial three days and subsequently stabilized. The glycoside hydrolases (GHs) and glycosyl transferases (GTs) emerged as two pivotal enzyme groups, comprising 4,837 and 1,882 genes, respectively. Notably, the hemicellulase and cellulase family genes within the GHs played a crucial role in breaking down hemicellulose and cellulose. A variety of lignocellulose-degrading enzymes were identified among the GHs. Under optimal conditions, the microbial system demonstrated a high efficiency in degrading corn straw, achieving degradation rates of 56.83% for lignin, 39.45% for cellulose, and 32.86% for hemicelluloses.

Conclusion

The microbial system domesticated by the restrictive culture method has a good ability to degrade lignocellulose of corn straw, which can further promote the process of corn straw feed.

Introduction

The remaining corn straw after crop harvest represents an important biomass resource (Sun et al., 2018). In China, the annual production of corn straw exceeds 1 × 1011 kg (Gu et al., 2025). However, a substantial portion of this biomass is directly burned, resulting in environmental pollution, resource wastage, loss of valuable nutrients, and increased anthropogenic carbon emissions (Benedetti et al., 2021). Although corn straw holds potential as a raw material for animal feed in animal husbandry, its primary component, lignocellulose, is a complex polymer and difficult to be converted by enzyme (Zoghlami & Paës, 2019). Lignocellulose in corn straw is mainly composed of 44.8%–44.9% cellulose, 26%–31% hemicellulose, and 19.9%–20.1% lignin (Wojcieszak et al., 2020). Cellulose has high degree of polymerization and highly ordered crystalline structure that makes it very resistant to enzymatic hydrolysis (Xu et al., 2020a; Xu et al., 2020b). It is composed of uniform glucose polymers with a degree of polymerization between 7,000 and 15,000 (Mehta & Chelike, 2024). Hemicellulose is a polymer without specific morphology, which forms a hard network structure in the cell wall together with cellulose (Lin & Lu, 2020). Lignin is a high molecular aromatic polymer formed by phenylpropane structural monomer connected by ether bond or carbon–carbon bond (Sun, Liu & Fu, 2019). It has a three-dimensional network structure, wrapped in the outer surface of cellulose to prevent enzyme contact. In addition, cellulase and other enzymes are irreversibly adsorbed during hydrolysis due to its hydrophobic structure (Zeng et al., 2014). The degradation of lignocellulose needs the synergetic effect of multiple enzymes. While, most non-ruminant animals lack the enzymatic capacity to digest and absorb such material, leading to a feed conversion efficiency of less than 20% (Kenneth & Brewster, 2017).

Microbial fermentation technology offers an effective solution to this problem. Microorganisms are capable of secreting cellulases, hemicellulases, and ligninases that degrade the lignocellulose content in straw. This not only enhances the digestibility of crude fiber and increases the crude protein content in corn straw but also improves its nutritional value, palatability, and digestibility for animals. Consequently, the utilization rate of corn straw is increased, animal growth is supported, and environmental sustainability is promoted (Wang & Yang, 2009; Chen et al., 2017). Additionally, the activity of lignocellulases produced through microbial fermentation and their degradation efficiency often surpass those of commonly used feed additives in China. Therefore, cultivating microorganisms that efficiently degrade lignocellulose and produce high-activity lignocellulases holds significant promise.

Compared to a single microbial strain that produces a single enzyme system, microbial consortia exhibit better adaptability and resilience in complex environments due to synergistic interactions among diverse microbial populations. This synergy enhances their overall lignocellulose degradation capacity. For example, Cui et al. (2021) investigated the degradation efficiency of rice straw using both single and composite strains, reporting a degradation rate of 73.3% with the constructed composite strain BD2W1G1, whereas the single strain achieved only 50.9%. Fermentation culture methods for microbial systems include the strain compounding method and the restrictive culture method. The strain compounding method has the limitations of high cost and long period, and needs to obtain a single strain from culturable microorganisms and clarify the synergistic mechanism within the flora (Xu et al., 2020a; Xu et al., 2020b). The restrictive culture method takes a short time and low cost, and efficiently enriches the natural microbial flora with strong adaptability by controlling the carbon source and temperature. Studies have demonstrated that microbial communities selected via restrictive culture methods display superior lignocellulose degradation activity. Zheng et al. (2020) employing a restrictive culture approach, continuously enriched strain LTF-27 under facultative anaerobic and static conditions at 15 °C for 20 days. This process resulted in the weight loss of cellulose, hemicellulose, and lignin in rice straw by 71.7%, 65.6%, and 12.5%, respectively (Zheng et al., 2020).

In order to expand the potential applications of corn straw, a microbial system specialized in lignocellulose degradation was cultivated using a static temperature increasing restricted culture method. The structural diversity and functionality of bacteria within this system were examined using high-throughput sequencing and metagenomics sequencing methods. This study aims to develop a microbial system capable of efficiently degrading lignocellulose in corn straw to enhance its use as fermented feed and to advance the utilization of corn straw as a sustainable feed resource.

Materials and Methods

Materials

Samples for strain screening were collected from the feces of pigs demonstrating robust growth, fed on peanut shells rich in lignocellulosic materials in Harbin, Heilongjiang, China. The pig manure was immediately transported to the laboratory in an ice pack and subsequently stored at 4 °C.

Corn straw was harvested from the Xiang Fang field in Harbin. The dried corn straw underwent a soaking process in 1% (w/v) NaOH for 24 h, followed by thorough rinsing with running water until a neutral pH was achieved. The pretreated corn straw was then dried at 105 °C, pulverized, and sieved through an 80-mesh screen. The processed corn straw was stored in a plastic bag at room temperature until required for use. Additionally, all chemicals employed were of analytical reagent grade and sourced from commercial suppliers.

Screening of microbial system

Using a restrictive cultivation method at an initial temperature of 30 °C, 5 g of the central portion of pig manure was added to 100 mL of peptone cellulose medium (PCS) on a clean bench with filter paper as the observation index (Pan, Zheng & Xiang, 2021). The PCS composition included 5% peptone, 5% cellulose, 5% NaCl, 0.1% yeast extract, 2% CaCO3, and 1% yeast powder. Once the filter paper fractured, 5 mL of the culture liquid was transferred to the aforementioned medium, and the fracture time was recorded. Microbial systems demonstrating weak degradation capabilities were eliminated using an incremental temperature gradient method, while the remaining microorganisms were preserved. Ultimately, cultures exhibiting high degradation ability and stability were selected as seed inocula and stored in 80% glycerol at −80 °C.

Identification of degradation ability

Congo red and aniline blue staining were employed to confirm the degradation capabilities of cellulose and in lignin of the microbial system (Putri & Setiawan, 2019). The microbial system was cultivated in Na carboxymethylcellulose (CMC) medium, which consisted of 1% sodium carboxymethylcellulose, 0.2% NaNO3, 0.05% MgSO4, 0.1% K2HPO4, 0.001% FeSO4, 0.05% KCl, and 2% agar, at 35 °C for three days. The medium was then stained with a 1 g L−1 Congo red solution for 30 min, followed by decolorization with a 1 g  L−1 NaCl solution for 20 min. The degradation capability for cellulose was determined by measuring and recording the diameters of the transparent circle and the colony, and comparing their ratio. The ratio of the transparent circle diameter (D) to the colony diameter (d) indicates the cellulose degradation ability. Similarly, the microbial system was cultured in aniline blue medium (10% yeast extract, 20% glucose, 0.1% aniline blue, and 20% agar) at 36 °C for seven days. Observations were made to determine whether the medium faded or a transparent circle appeared. The diameter ratio was measured to assess the potential for degrading lignin.

Determination of degrading enzyme activity

1% microbial system was cultured in two distinct media: cellulose enzyme-producing medium (1% sodium carboxymethylcellulose, 0.4% (NH4)2SO4, 0.2% KH2PO4, 0.05% MgSO4, 1% peptone, 0.5% beef extract) and lignin enzyme-producing medium (10% peptone, 3% yeast extract, 20% sucrose, 0.5% NaCl), both at 180 rpm and 36 °C for seven days. Media without inoculum served as blank controls. Enzymatic activities, including Filter Paper Activity (FPA), Exoglucanase (C1), Endoglucanase (CX), and β-glucosidase, were quantified using 3,5-Dinitrosalicylic acid (DNS). Lignin peroxidase (Lip) activity was measured using veratrol at 310 nm, Mn-dependent peroxidase (Mnp) activity at 240 nm, and Lactoperoxidase (Lac) activity was determined by 2,2’-Azinobis-(3-ethylbenzthiazoline-6-sulphonate) (ABTS) at 420 nm (Putri & Setiawan, 2019).

Analysis of degradation characteristics

Determination of OD600, pH, volatile fatty acid, soluble sugar and total sugar

5% microbial system was cultured in PCS medium for 7 days, during which broth samples were taken at predetermined intervals to assess various parameters. After fermentation, the broth was centrifuged at 12,000 rpm for 5 min. The supernatant was then discarded, and the pellet was resuspended in sterile water. The optical density at 600 nm (OD600) was measured, and a growth curve was plotted based on these values. pH levels were determined using a pH meter. Volatile fatty acids (VFAs) were analyzed using a High Performance Liquid Chromatograph (HPLC) U3000 (Thermo Fisher Scientific), equipped with an ultraviolet detector set at 214 nm. The mobile phase consisted of 100 mM sodium sulfate, with a flow rate of 1 mL min−1. The column temperature was maintained at 30 °C, the injection volume was 10 µL, and the pH was adjusted to 2.5 using methanesulfonic acid. Lactic acids were quantified using an Acclaim Polar Advantage II C18 Column (50 mm × 4.6 mm, 3 µm), while other fatty acids were analyzed using an Acclaim Organic Acid LC Column (250 mm × 4 mm, 5 µm). Soluble sugars (Sc) and total sugars (Ts) in the broth were determined respectively using the anthrone colorimetry method and the phenol sulfuric acid method (Qing et al., 2021).

Determination of corn straw, cellulose, hemicellulose and lignin degradation rates

To determine the degradation rates of corn straw, cellulose, hemicellulose, and lignin, the weight-loss method combined with the Van Soest protocol was employed (Van Soest, Robertson & Lewis, 1991). The microbial system was inoculated into a fermentation medium containing 2% corn straw powder, 2% KH2PO4, 0.5% MgSO4, 4% (NH4)2SO4, 10% peptone, and 5% beef extract. Samples were taken over a 7-day period. The degraded corn straw was filtered using constant weight filter paper with mass m0, and any soluble substances were removed by washing with water. The residues of corn straw were then dried at 105 °C for 1 h and allowed to stand in a desiccator for 30 min. These residues were weighed, with the initial weight recorded as m1. The drying and weighing steps were repeated until the sample reached a constant weight, denoted as m2. The final weight of the residue, m3, was calculated as m2-m0. m was the initial dry weight of the corn straw before inoculation. The formula to calculate the degradation rate of corn straw is: Y=m−m3m×100%.

Microbial system diversity analysis

During the investigation of the lignocellulose degradation characteristics by the microbial system, samples were taken at designated periods: the initial (0 and 24 h), peak (48 and 72 h), and end (96 and 120 h). Under sterile conditions, microbial samples from different degradation periods were collected and centrifuged at 12,000 rpm for 10 min. Total DNA was extracted from the microbial system using the Fast DNA Spin Kit for Soil. The quality and concentration of the extracted DNA were assessed by spectrophotometry and agarose gel electrophoresis. Each period included three replicated experiments, and the obtained samples were cryopreserved with dry ice and sent to Wuhan Fisha Gene Information Co., Ltd. for analysis. For microbial community analysis, the V3–V4 regions of the 16S rRNA genes were amplified using primers 338 F (ACTCCTACGGGAGGCAGCAG) and 806 R (GGACTACHVGGGTWTCTAAT). The amplified products were then purified and quantified. The resulting sequences underwent double-end sequencing on the Illumina MiSeq platform. High-quality sequences were clustered into Operational Taxonomic Units (OTUs), and species classification was performed using the Feature tool. To analyze microbial diversity, Alpha diversity index was employed to assess species diversity within the same sample, whereas Beta diversity analyses were used to compare species diversity across different samples from various degradation periods.

Functional microorganism analysis

The microbial system cultured on PCS was collected during its peak period for metagenomic sequencing analysis. Total genomic DNA was extracted using the OMEGA DNA Spin Kit for Soil. The extracted DNA then underwent a series of processing steps for metagenomic sequencing. To enhance the quality and reliability of the sequencing data, flanking sequences and nitrogenous bases with a mass below 3 at the double end were trimmed using Cutadapt (v1.2.1). Quality trimming continued with the scanning of reads using a sliding window of 4 bp, cutting when the average quality score dropped below 15. In instances of contamination by host DNA, Bowtie2 was employed to filter out host-related reads. The high-quality, filtered sequences were then assembled into contigs using the megahit software (v1.1.2). These contigs were further analyzed with MetaGeneMark-v3.38 to predict genes, discarding sequences shorter than 100 nucleotides. To minimize redundancy, the predicted results from different samples were consolidated using CD-HIT-v4.8.11, clustering them at 95% identity and 90% coverage parameters. The non-redundant gene catalog was then compared against Clean Data using Bowtie2 to ensure accuracy. Finally, the reads mapping to each gene were quantified to ascertain gene abundance in each sample.

Changes in morphology, functional groups and essential components of corn straw

The effects of carbon source type, carbon source addition level, nitrogen source type, nitrogen source addition level, temperature, fermentation duration, and inoculation amount on the degradation rate of corn straw were investigated using a single-factor rotation method. Optimal degradation conditions were identified through this analysis. Under these optimal conditions, the microbial system for corn straw degradation was subjected to fermentation and cultivation using a static temperature-increasing approach combined with a restrictive cultivation method. To analyze the structural changes in the corn straw, samples from before (control group) and after degradation (experimental group) were examined using scanning electron microscopy (SEM) and Fourier transform infrared spectroscopy (FTIR) (Gong et al., 2020). These methods provided insights into the morphological changes and alterations in functional groups due to microbial action. The degradation rates of cellulose, hemicellulose and lignin in the corn straw were quantitatively assessed using the Van Soest method (Van Soest, Robertson & Lewis, 1991). The calculation of degradation rates was based on the following formula: Y=m−m′m×100% m: weight before degradation; m’: weight after degradation.

Data and bioinformatic analysis

The data for enzyme activity, OD, pH, VFAs, sugar content, and lignocellulose degradation rate were analyzed using Origin 2019b and IBM SPSS Statistics 23. The degree of dispersion of data was highlighted using standard deviation error bars. STDEV.S function in Excel 2024 was used to calculate the standard deviation. Duncan’s new multiple range test was used for the post-hoc test of Analysis of Variance (ANOVA). Different letters indicate statistically significant differences (p < 0.05). 16S rRNA data analysis was performed using the Majorbio Cloud (http://www.majorbio.com). Kruskal–Wallis test was used to analyze the difference of α diversity index between groups (krus.test function in R (version 3.3.1), 0.001 < p < 0.01 marked as **, p ≤ 0.001 marked as ***). Principal component analysis (PCA) was used to evaluate the differences in bacterial community structure according to Bray-Curtis distance and Orthogonal Projections to Latent Structures-Discriminant Analysis (OPLS-DA) in R. The blastp command in DIAMOND software (v2.0.9) was employed to align protein sequences of Unigenes against the full KOBAS (v3.0) database. KOBAS3 software was then used to analyze the alignment results (Zhang et al., 2018). Functional annotation at various hierarchical levels was performed using the relationship tables available in the Kyoto Encyclopedia of Genes and Genomes (KEGG, http://www.genome.jp/kegg) database. The abundance of each functional category was calculated by summing the gene abundances corresponding to KEGG Orthology (KO), pathway, Enzyme Commission (EC), and module identifiers. Carbohydrate-active enzymes were annotated using the bCAN-HMMab-V10 database within dbCAN2 (Xie et al., 2011) (https://bcb.unl.edu/dbCAN2_obsolete/), with sequence alignment performed using HMMER v3.3.2. Raw sequence data have been deposited in the National Center for Biotechnology Information (NCBI) database under accession number PRJNA1262545.

Results

Screening of microbial system

After 15 groups of the microbial system were sub-cultured across 50 generations using static temperature increasing and restrictive cultivation methods, when the temperature was 36 °C, the breaking days of each generation of indicator were 3 days, and then hold steady, and the obtained microbial system was preliminarily recognized as having the ability to degrade lignocellulose and reached a relatively stable state.

The Congo red staining results were particularly striking, with a diameter ratio (D/d) of 19.76 (Fig. 1A) and aniline blue staining further supported these findings with a D/d ratio of 38 (Fig. 1B).

Figure 1 Congo red (A) and aniline blue staining results (B).

Enzyme activity of microbial system

The established correlation between glucose content and absorbance value demonstrated a good fit within the range of 0.2–1.0 mg mL−1, making it a reliable basis for calculating enzyme activity. Enzyme activity serves as a critical metric for assessing microbial activity and the effectiveness of lignocellulose degradation.

The recorded activities for Filter Paper Activity (FPA), Exoglucanase (C1), Endoglucanase (CX), and β-glucosidase were 33.21 U mL−1, 36.34 U mL−1, 70.6 U mL−1, and 52.4 U mL−1, respectively (Fig. 2A). Similarly, the degradation of lignin is influenced by the activities of Lignin Peroxidase (LiP), Laccase (Lac), and Mn-dependent Peroxidase (MnP), with their activities measured at 485 U L−1, 376 U L−1, and 194 U L−1, respectively (Fig. 2B).

Figure 2 The result of cellulase activity (A) and lignin enzyme activity (B).

Different letters indicate statistically significant differences (p < 0. 05).

Degradation performance of microbial system

OD600values

Initially, there was a rapid increase in the OD600 value from 0.18 to 0.81 over the first three days. After the 3rd day, the increase in OD600 value slowed, reaching a peak of 0.96 on the 5th day (Fig. 3A). By the 7th day, the OD600 value decreased to 0.7.

Figure 3 The result of OD value change (A), pH change (B), volatile fatty acid changes (C), soluble sugar content change (D), total sugar content change (E) and degradation rate of corn straw and its main components (F).

Different letters indicate statistically significant differences (p < 0.05).

pH values

Initially, the pH values dropped sharply from 8.8 to 8.2 on the 2nd day as lignocellulose was degraded. Subsequently, the pH began to rise slowly, returning to 8.5 by the 5th day, and then stabilized until the end of fermentation (Fig. 3B).

Volatile fatty acids

The concentrations of key VFAs increased rapidly during the first three days, peaking on the 3rd day with concentrations of 775.68 mg L−1, 368.15 mg L−1, 390.32 mg L−1, and 778.88 mg L−1 for lactic acid, acetic acid, propionic acid and butyric acid, respectively. Notably, lactic acid and butyric acid exhibited the highest concentrations (Fig. 3C).

Soluble sugar

During the lignocellulose degradation process, the concentration of Sc peaked on the 3rd day, rising from 3.8 to 8.9 µg mL−1, before slowly decreasing and eventually stabilizing. This trend is depicted in Fig. 3D.

Total sugar

Ts, an intermediate product of cellulose degradation, decreased rapidly from 4.0 mg mL−1 to 1.6 mg mL−1 within the first three days and then stabilized (Fig. 3E).

Lignocellulose degradation rates

The degradation rates of corn straw, cellulose, hemicellulose, and lignin increased initially and then stabilized, peaking on the 3rd day. After 72 h of fermentation, the degradation rates were 31.49% for corn straw, 37.34% for cellulose, 28.62% for hemicellulose, and 53.84% for lignin (Fig. 3F).

Structural of microbial communities

α-diversity of bacterial community

From Chao1, Shannon, observed species, and Faith PD indexes, we can found that Each index initially increased and then decreased, with the highest levels of bacterial community abundance and species count observed at 48 h, closely followed by the results at 72 h (Fig. 4).

Figure 4 Alpha diversity index of bacterial community in microbial system.

(A) Chao1 index indicates the community richness. (B) Shannon index indicates the community diversity. (C) Observed species index indicates number of species. (D) Faith pd index indicates phylogenetic diversity. The asterisk denotes statistically significant differences **0.001 < p < 0.01; ***p ≤ 0.001.

β-diversity of bacterial community

The groups at 0, 24, 96, and 120 h showed relatively close distances in PCA space (Fig. 5A). In contrast, the distances between the groups at 48 h and 72 h were significantly greater, suggesting substantial differences in species composition during these periods. This was consistent with the results from the Unweighted Pair Group Method with Arithmetic Means (UPGMA), which also indicated that the microbial systems at 0, 24, 96, and 120 h clustered into an independent branch, showing high similarity and uniformity, while the groups at 48 h and 72 h were distinctly different (Fig. 5B).

Figure 5 Beta diversity index of bacterial community in microbial system.

(A) PCA presenting differences in microbial community structure. (B) UPGMA presenting similarity and difference of community structure.

Species numbers generally increased initially and then decreased over time (Fig. 6A). The total number of species across different degradation periods was 291, peaking at 1,215 at 48 h, followed by 1,034 and 917 at 24 and 72 h, respectively. The microbial system predominantly comprised phyla such as Proteobacteria, Bacteroidetes, Firmicutes, Fibrobacteres, and Spirochaetes (Fig. 6B). Further investigation revealed that Pseudomonas, Fibrobacter, Macellibacteroides, Sedimentibacter, Bacteroides, Comamonas, Brevibacillus, and Treponema collectively accounted for 50–82% of the relative abundance of the microbial system at the genus level across different degradation periods (Fig. 6C).

Figure 6 Analysis of species differences and diversity in microbial systems.

(A) Flower plot presenting the number of species in different periods. (B) Relative abundance of the top five most abundant phyla. (C) Relative abundance of the top fifteen most abundant genus.

Carbohydrate-active enzymes (CAZy) functional annotations

To investigate the lignocellulose degradation capability of the microbial system, the CAZy enzyme database was utilized to analyze the diversity of carbohydrate-active enzymes at 48 and 72 h. Six protein modules were annotated from the database: Auxiliary Activities (AAs), Glycosyl Hydrolases (GHs), Glycosyl Transferases (GTs), Polysaccharide Lyases (PLs), Carbohydrate Esterases (CEs), and Carbohydrate-Binding Modules (CBMs). Among these, GHs and GTs were the dominant groups with 4,837 and 1,882 genes, respectively (Fig. 7A). The abundance of GHs was 55% at 48 h and 51% at 72 h, while the abundance of GTs was 21% at 48 h and 18% at 72 h (Figs. 7B and 7C). The top 30 gene families identified included GT4, CBM6, CE1, GT51, GH23, GH3, GH2, GH9, GH109, GH20, CBM35, CE4, GH92, GT28, GH8, CBM4, GT9, PL1, GH171, GH57, GH18, GH97, CE11, GT19, GH10, CE6, GH16_3, GH5_2, GT30, GH733-11 (Fig. 7D).

Figure 7 Examples of CAZy fermentation for 48 and 72 h in microbial system.

(A) The result of carbohydrate activity enzymes annotation. (B) The proportion of CAZy annotation abundance in microbial system fermented for 48 h. (C) The proportion of CAZy annotation abundance in microbial system for fermented 72 h. (D) The results of the first 30 CAZy carbohydrate active enzyme genes.

Functional predictions

The potential functional analysis of bacterial communities associated with lignocellulose degradation was visualized in Fig. 8. Using the Phylogenetic Investigation of Communities by Reconstruction of Unobserved States (PICRUST) and the Kyoto Encyclopedia of Genes and Genomes (KEGG) database, the ecosystem functions of the bacterial community were assessed. Figure 8A displays the proportion of Metabolism (Me), Genetic Information Processing (Gp), Environmental Information Processing (Ep), Cellular Processes (Cp), Human Diseases (Hd), and Organismal Systems (Os) in the KEGG pathways during microbial system fermentation at 48 and 72 h, with Me and Gp showing higher proportions and relative abundance accounting for 22% of the relative abundance. The potential functions of bacteria at 48 and 72 h, depicted in Figs. 8B, 8C, indicate that functional genes primarily participate in amino acid metabolism, carbohydrate metabolism, cofactors and vitamin metabolism, energy metabolism, and signal transduction, with a relative abundance ranging from 19% to 21%. During the thermophilic stage, a significant portion of the bacterial community contributes to the conversion of lignocellulose and proteins.

Figure 8 Functional prediction results of microbial system fermentation for 48 and 72 h.

(A) Primary metabolism of KEGG pathway in microbial system fermentation for 48 and 72 h. (B) Secondary metabolism of KEGG pathway in microbial system fermentation for 48 and 72 h. (C) Tertiary metabolism of KEGG pathway in microbial system fermentation for 48 and 72 h.

SEM observations

The optimal conditions for degradation were determined to be lactose as the carbon source at a concentration of 2%, ammonium nitrate as the nitrogen source at 4%, and an inoculation rate of 5%. Under these conditions, with a fermentation temperature of 36 °C and a duration of 3 days, the corn straw degradation rate reached 36.92%.

SEM observations of corn straw before and after degradation under optimal conditions were displayed in Fig. 9. Initially, the surface structure of the straw was complete, orderly, and smooth. Post-degradation, the surface showed significant damage; holes, cracks, and areas of fracture and collapse were evident (Figs. 9A and 9C). The protective wax-silicified layer and the outer parenchyma tissue had disappeared, exposing the inner parenchyma tissue (Dar et al., 2021) (Figs. 9B and 9D). These SEM observations suggest that the microbial and enzyme systems were able to penetrate the corn straw, resulting in a looser structural integrity. The outer thin wall of the straw was almost completely degraded, revealing numerous cracks and voids. The wax silicified layer thinned significantly, and the lower epidermis weakened due to the mycelial growth and the secretion from the cell enzyme system.

Figure 9 SEM observation of straw before (A) and after (B) degradation (200x) and SEM observation of straw before (C) and after (D) degradation (1,000x).

FTIR observation

FTIR observation of straw before and after degradation at wavelengths ranging from 500 to 4,000 cm−1 was depicted in Fig. 10. The stretching peak of O-H, predominantly at 3,440 cm−1, appeared significantly weaker in the experimental group compared to the control group, suggesting potential cellulose degradation. The transmittance at 1,788 cm−1 was noticeably higher in the experimental group than in the control group. The carbon-oxygen double bond in ketone and carboxylic ester compounds vibrated at 1,740–1,720 cm−1. The carbon–carbon double bond in aromatic structures, amides, ketones, or quinones was observed at 1,610–1,680 cm−1. The benzene ring of aromatic compounds vibrated at 1,510–1,500 cm−1. Si-O or C-O in polysaccharides was detected at 1,080–1,040 cm−1 (Baz et al., 2018). The significant alterations in the relative intensity and shape of absorption bands at 3,440 cm−1, 1,788 cm−1, 1,725 cm−1, 1,657 cm−1, 1,500 cm−1, and 1,044 cm−1 after degradation suggest changes in the functional groups related to cellulose, hemicellulose, and lignin. This confirms the microbial system’s capability to degrade straw effectively.

Figure 10 FTIR of straw before and after degradation.

Main components of straw before and after degradation under the optimal conditions

A variety of microorganisms within the microbial system synergistically enhance the production of extracellular enzymes in the fermentation system, facilitating straw degradation. After the microbial system was inoculated, the degradation rates of cellulose, hemicellulose, and lignin in straw were recorded at 39.45%, 32.86%, and 56.83%, respectively. This indicates a notably stronger capability of the microbial system to degrade lignin compared to cellulose and hemicellulose (Fig. 11).

Figure 11 Change of main composition of straw before and after degradation under the optimal conditions.

Discussion

As a major agricultural country, China produces a substantial amount of corn straw, a large portion of which is burned in situ. This practice can lead to the loss of essential nutrients such as nitrogen, phosphorus, and potassium required for crop growth, ultimately contributing to soil degradation. Additionally, the soil hosts a diverse community of beneficial microorganisms that exist symbiotically with crops. The high temperatures generated by straw incineration disrupt these beneficial microbial communities and damage the ecological environment. Moreover, incineration destroys the soil aggregate structure, significantly impairing the soil’s ability to retain water and nutrients, thereby reducing its fertility and sustainability (Bi, Wang & Gao, 2009). The application of microbial fermentation technology for the treatment of corn straw can effectively convert it into high-quality animal feed, thereby maximizing the utilization of corn straw resources and mitigating the aforementioned detrimental effects on soil health and the broader ecosystem.

Restrictive culture microbial systems to degrade lignocellulosic materials is one of the technologies developed in recent years, which can maintain the synergistic relationship between species in natural populations. In this study, limited subculture was used to select efficient and stable microbial systems with lignocellulosic degradation ability from pig manure. And the microbial system shows significantly higher efficiency in lignocellulosic degradation. When compared to previously reported cellulase-producing stains with D/d ratios typically ranging from 1.2–3.1 and 1.40–2.18, the results from this study are significantly higher, exceeding those values by factors of 6 and 9, respectively (Putri & Setiawan, 2019; Saha, Roy & Hossen, 2021). The efficacy of aniline blue staining in this study was 19 times greater than that reported in previous studies (Li et al., 2020a; Li et al., 2020b; Li et al., 2020c). Liu et al. (2022a); Liu et al. (2022b) employed a single-strain compounding method to construct a complex microbial community, in which the activities of filter paperase (FPA), C1, carboxymethyl cellulase (CX), and β-glucosidase were 24.12 U mL−1, 25.31 U mL−1, 55.80 U mL−1, and 49.13 U mL−1, respectively. The cellulase activity of the microbial system used in this study was higher than those reported by Liu et al. Notably, the activity of the CX enzyme in this microbial system significantly surpasses that of single strain, such as the soil-derived strain Streptomyces DSK59 and the paper mill-derived strains Arthrobacter sp. AXJ-M1 and Serratia sp. AXJ-M (An et al., 2022; Budihal, Agsar & Patil, 2016). Xu et al. (2021) screened the lignin-degrading bacterial strain LDC, which exhibited lignin peroxidase (Lip) activity of 405.26 U L−1, manganese peroxidase (Mnp) activity of 427.98 U L−1, and laccase (Lac) activity of 143.98 U L−1. The lignin-degrading enzyme activities of the microbial system in this study were higher than those reported for strain LDC. Furthermore, the activities of Lip and Lac in this microbial system are approximately ten times higher than those reported for Aspergillus Flavus F-1 isolated from soil (Li et al., 2020a; Li et al., 2020b; Li et al., 2020c). The diameter of the aniline blue transparent zone is indicative of lignin peroxidase (Lip) enzyme activity. The results demonstrated that Lip exhibited the highest activity among the lignin-degrading enzymes, suggesting that the enzyme activity data were consistent with the aniline blue staining observations. The domesticated microbial system exhibited the capacity to degrade both cellulose and lignin, with lignin degradation activity exceeding that of cellulose. The enhanced activity of cellulase and lignin enzymes may be attributed to the continuous release of hydrolases under synergistic effects facilitated by sufficient nutrients, high cell concentration, and vigorous metabolism in the microbial system (Yan et al., 2022).

The degradation characteristics of the microbial system were analyzed to evaluate its capacity for corn straw degradation. Changes in optical density (OD) values may be attributed to nutrient depletion in the fermentation broth, increased interspecific and intraspecific competition, reduced bacterial concentration, and decreased metabolic activity due to intensified competition for nutrients and space. As culture time increased, the microbial system transitioned from the logarithmic growth phase to the stationary and decline phases, consistent with the characteristic stages of a microbial growth curve. The observed changes in OD values were consistent with those reported for the microbial consortium LDC (Xu et al., 2021). Variations in OD values in microbial systems are closely related to microbial concentration (Benner et al., 2020). In this study, the rapid increase in OD value during the early stages of cultivation indicates a swift rise in microbial concentration, which supports the effective degradation of corn straw. During anaerobic fermentation, microorganisms metabolize substrate nutrients to produce key VFAs such as lactic acid, acetic acid, propionic acid, and butyric acid (Sarma et al., 2022). The observed trend in VFA concentrations inversely correlates with the changes in pH values. This relationship is due to the production of small molecules such as amino acids, peptides, fatty acids, glycerol, and Sc through the degradation of lignocellulose by the microbial system. These small molecules are initially acidified by acid-producing microorganisms into organic acids, which are subsequently converted into VFAs through acetoxylation processes. Due to the buffering capacity of calcium carbonate in the fermentation broth and the metabolic regulation by alkali-producing microorganisms, the pH of the fermentation broth did not decrease significantly (Benner et al., 2020). As the degradation process progressed, the accumulation of organic acids decreased, and the presence of bacteria within the microbial system capable of utilizing organic acids as nutrients contributed to the gradual recovery of pH in the microbial system. Different microbial groups contribute distinctively to VFA production. For instance, bacteria within the Firmicutes phylum can ferment carbohydrates to produce lactic acid, whereas Bacteroidetes can decompose peptone or glucose to yield various VFAs, including succinic acid, acetic acid, formic acid, lactic acid, and propionic acid (Silva et al., 2022). These findings indicate that the microbial system effectively produces VFAs, with the fermentation of corn straw predominantly occurring through lactic and butyrate fermentation pathways. In general, the fermentation broth of the microbial system exhibits an alkaline pH, primarily due to the presence of a diverse consortium of microorganisms and the abundance of nitrogen-containing organic matter in pig manure, which serves as a nitrogen source for adjusting the carbon-to-nitrogen (C/N) ratio (Wang et al., 2022). Additionally, the production of NH4+-N during the generation of organic acids further contributes to maintaining the pH of the microbial system at an alkaline level. Sc serves as a crucial intermediate product in the conversion from organic substances to VFAs (Sarma et al., 2022). The concentration of Sc is closely linked to both the pH levels and the microbial system’s efficiency in degrading cellulose and transforming it into VFAs. The variation trends of Sc and VFAs correlate positively, showing a simultaneous increase and subsequent stabilization, which is inversely related to the changes in pH value. Initially, cellulose is broken down into Sc by the microbial system, primarily to provide energy for microbial activities (Karlsson et al., 2012; Karlsson et al., 2013). These Sc are then converted into VFAs. However, when the microbial concentration is low, this conversion process becomes less efficient, leading to a stabilization in the concentration of Sc as insufficient microorganisms converting all available Sc into VFAs. This behavior underscores the dynamic interplay between microbial activity, substrate degradation, and product formation within the system, highlighting the importance of maintaining adequate microbial concentrations for optimal bioconversion processes. The microbial system gradually adapted to the conditions within the medium and effectively absorbed nutrients, thereby enhancing interspecific synergistic interactions, which led to an increase in the degradation rate. However, as interspecific competition within the system intensified and nutrient availability declined, the microbial population density decreased, metabolic activity slowed, enzymatic reaction rates diminished, and the straw degradation rate eventually decreased and stabilized at a constant level. Wang et al. (2022) employed piggery wastewater to domesticate microorganisms derived from paddy soil, resulting in a composite microbial system that achieved degradation rates of approximately 50% for cellulose, 20% for hemicellulose, and 10% for lignin. Chen et al. (2025) utilized yeast strains expressing lignin peroxidase (Lip) and manganese peroxidase (Mnp) to co-ferment wheat straw in combination with other cellulase- and hemicellulase-producing strains, achieving a lignocellulose degradation rate of 45.42%. These results indicated that the domesticated microbial system potentially possesses superior degradation abilities, with a notably stronger capability for lignin degradation compared to cellulose degradation.

Elucidation of functional microorganisms will aid in explaining the underlying mechanisms and characteristics of lignocellulose degradation (Marynowska et al., 2020). The composition of this microbial community was predominantly comprised with Proteobacteria, Bacteroidetes, Firmicutes, Fibrobacteres, and Spirochaetes. Shen et al. (2022) investigated an ectopic fermentation system utilizing pig manure and straw waste and identified Proteobacteria, Firmicutes, Bacteroidetes, and Actinobacteria as the dominant bacterial phyla. Similarly, other domesticated systems using rice straw as the substrate, such as XDC-2, WSD-5, and MC1 (Wang et al., 2013; Wen et al., 2012; Yu et al., 2016), exhibited dominant bacterial communities comparable to those observed in this study. Compared to XDC-2, the microbial composition in this study is more complex. Proteobacteria, facultative anaerobes, can degrade propionic and butyric acids and utilize various carbon sources, converting sulfuric acid to hydrogen sulfide, which is beneficial for intestinal health. Bacteroidetes can hydrolyze and acidify protein into small molecular organic acids, and hydrolyze cellulose and polysaccharides, involving in carbohydrate metabolic pathways (Hess et al., 2011). Firmicutes utilize nitrogen sources from pig manure, enhancing amino acid transport and metabolism (Dai et al., 2012). Fibrobacteres are known for their ability to degrade cellulose, while Spirochaetes contribute to hemicellulose degradation and interact with cellulose-degrading bacteria. Studies have shown that the dominant bacterial phyla in the gastrointestinal tract of pigs are Firmicutes and Proteobacteria, which is consistent with the results of this experiment. This indicates that the domestication process of the microbial system does not alter the dominant microbial communities present in pig manure (Hu et al., 2025). At the genetic level, Pseudomonas, Brevibacillus, Treponema and Comamonas were the dominant genera. Pseudomonas supports low oxygen environments conducive to anaerobic bacterial activity and possesses strong lignin degradation capabilities. Brevibacillus, primarily aerobic, metabolizes sugars into acids (Shida et al., 1996). Treponema is associated with the digestion of plant-derived carbohydrates, and Comamonas degrades aromatic compounds within organic acid and amino acid nutrient systems (Cai et al., 2018).

Amino acids synthesized via metabolic processes serve as nitrogen sources for microorganisms, supporting the synthesis of proteins, peptides, and other nitrogen-containing compounds. α-Keto acids generated through deamination and transamination reactions are involved in synthesizing non-essential amino acids, transforming into sugars and esters, and are oxidized to produce energy (Kanokratana et al., 2018). The carbon dioxide produced during glycolysis and the tricarboxylic acid cycle increases environmental CO2 concentration, which is advantageous for the growth of facultative anaerobic bacteria. Overall, the carbohydrate, cofactor, and energy metabolisms are crucial for providing the essential nutrients needed for the growth and reproduction of bacteria (Liu et al., 2022a; Liu et al., 2022b).

The results of SEM of straw before and after degradation were consistent with the findings reported by Gong et al. (2020); Xu et al. (2016). These observations support the transformation of organic silicon into inorganic silicon, the conversion of insoluble silicon into soluble silicon, and the gradual disappearance of the silicide layer. The microbial system effectively promotes the degradation of lignocellulose. Ma et al. (2020) reported degradation rates of cellulose, xylan, and acid-insoluble lignin by the thermophilic denitrifying bacterium Y7 as 18.64%, 12.96%, and 17.21%, respectively. Li et al. (2020a); Li et al. (2020b); Li et al. (2020c) found that the degradation rates of cellulose, hemicellulose, and lignin in corn straw by ten different strains were 44.4%, 34.9%, and 39.2%, respectively. Jin, Ai & Dong (2022) demonstrated that compounds in wheat straw, including cellulose, hemicellulose, and lignin, were significantly degraded after pretreatment, with degradation rates of 37.47%, 46.96%, and 14.05%, respectively. The microbial system developed in this study exhibited a strong lignocellulose degradation capacity, contributing effectively to the breakdown of straw.

Conclusion

The microbial system developed in this study demonstrated an efficient capacity to degrade lignocellulose of corn straw. This efficiency is attributed to the dominance of specific microbial phyla and genera, along with abundant enzymes that possess strong degradation capabilities. The primary phyla within the microbial system included Proteobacteria, Bacteroidetes, Firmicutes, Fibrobacteres, and Spirochaetes. Within this microbial ensemble, the gene families of Glycosyl Hydrolases (GHs) and Glycosyl Transferases (GTs) were particularly abundant and played a significant role in degradation processes. The SEM analysis revealed that the surface structure of the degraded straw was extensively damaged, exhibiting breakage and collapse. Additionally, FTIR characterization indicated alterations in the functional groups associated with lignocellulose. In terms of degradation performance under optimal conditions, the degradation rates for lignin, cellulose, and hemicelluloses in corn straw were 56.83%, 39.45%, and 32.86%, respectively. These results significantly advance the processing of lignocellulosic resources in corn straw, highlighting the effectiveness of the domesticated microbial system in breaking down complex biomaterials. Although the degradation of corn straw by domesticated microbial systems has achieved certain theoretical advances and these systems exhibit high lignocellulose degradation capacity, the practical application of straw degradation should be further expanded. In the future, metabolomics can be employed to investigate the small-molecule compounds generated through the synergistic degradation of lignocellulose by microbial systems. These compounds can then be analyzed in conjunction with predictive metabolic pathways to infer potential degradation mechanisms. Furthermore, microbial systems may be applied to ferment straw for the production of fermented feed, thereby advancing the utilization of corn straw as a sustainable feed resource.

Supplemental Information

Supplemental Information 1 Congo red and aniline blue staining results

Supplemental Information 2 Degradation characteristics

Supplemental Information 3 Structural of microbial communities

Supplemental Information 4 SEM observations

Supplemental Information 5 SEM observations

Supplemental Information 6 SEM observations

Supplemental Information 7 FTIR observation

Supplemental Information 8 Degradation rate data for Fig. 11

Additional Information and Declarations

Competing Interests

Author Contributions

Data Availability

The authors declare there are no competing interests.

Jiang Yu conceived and designed the experiments, performed the experiments, analyzed the data, prepared figures and/or tables, authored or reviewed drafts of the article, and approved the final draft.

Chun Wang conceived and designed the experiments, performed the experiments, analyzed the data, prepared figures and/or tables, authored or reviewed drafts of the article, and approved the final draft.

Yuena Qi conceived and designed the experiments, performed the experiments, analyzed the data, prepared figures and/or tables, authored or reviewed drafts of the article, and approved the final draft.

Jie Lian conceived and designed the experiments, performed the experiments, analyzed the data, prepared figures and/or tables, authored or reviewed drafts of the article, and approved the final draft.

Zhenhua Yu conceived and designed the experiments, authored or reviewed drafts of the article, and approved the final draft.

Guoquan Fan conceived and designed the experiments, authored or reviewed drafts of the article, and approved the final draft.

Jianming Sun conceived and designed the experiments, authored or reviewed drafts of the article, and approved the final draft.

Xitao Liu conceived and designed the experiments, authored or reviewed drafts of the article, and approved the final draft.

Xiaofei Liu conceived and designed the experiments, authored or reviewed drafts of the article, and approved the final draft.

The following information was supplied regarding data availability:

High-throughput sequencing sequences are available at NCBI: PRJNA1262545 and at figshare: Wang, Chun (2025). Investigating the acclimation and degradation characteristic of the microbial system in corn straw. figshare. Dataset. https://doi.org/10.6084/m9.figshare.28803563.v4.

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
