# Peer review of "Acclimation and degradation characteristic of the microbial system in corn straw"

_PeerJ, doi:10.7717/peerj.20386_

## Round 0.1 · original submission · Major Revisions

· Academic Editor

Major Revisions

Dear Dr. Wang, I ask you to carefully respond to the reviewers' fundamental comments. It is necessary to clearly describe to the readers of the article the relevance of your research.

**Language Note:** The review process has identified that the English language must be improved. PeerJ can provide language editing services - please contact us at [email protected] for pricing (be sure to provide your manuscript number and title). Alternatively, you should make your own arrangements to improve the language quality and provide details in your response letter. – PeerJ Staff

·

Basic reporting

This manuscript describes how an easy-to-domesticate microbial system breaks down lignocellulosic corn straw. The work is well-arranged, supplying all the important background data as well as plenty of references to publications in the area to better explain the findings. The outcomes are shown plainly with supporting graphics and tables, which demonstrate how much lignin, cellulose, and hemicellulose can be removed during degradation. The authors discuss the topic fully, uniting microbial activity and enzyme creation with the proposed hypotheses and concluding with an analysis of the applications of biomass processing.

Experimental design

It uses an appropriate design that fully relates to the principles and interests of the journal to investigate a significant gap in microbial breakdown of lignocellulosic materials. The research question is straightforward, asking how a domesticated microbial system can break down corn straw, making it important and meaningful for biomass processing. The ways of performing experiments are well-explained, so others can repeat them, and all work follows both technical and ethical guidelines. By taking a complete approach, the findings can offer meaningful information on bacterial metabolism and its impact on waste management and bioenergy.

Validity of the findings

While the findings in this study are right and have an important place in lignocellulosic degradation, their influence and novelty are not always clear. Talking about why replication matters and promoting well-planned studies will strengthen what is written in the current literature. The results are supported by good statistics, but more precise details of the experimental controls should be added for thorough confirmation. On the whole, the conclusions match the research question and are well presented; even so, the reader could appreciate more discussion on the value of the findings for real-world situations.

Additional comments

It provides useful evidence on how domesticated microbial systems manage the degradation of lignocellulosic biomass, showing much promise for use in bioenergy and waste management. Still, more information on soil health and the effects this microbial system has on ecosystems would make the discussion richer. In addition, comparing the results to what is known in other kinds of microbes could make the findings clearer. Enhancing the manuscript by describing the scalability of the proposed solutions for use in industrial systems will provide extra benefits and direct possible future work.

·

Basic reporting

Comments
The manuscript titled “Investigating the acclimation and degradation characteristics of the microbial degradation system in corn straw” aims to enhance the degradation rate of the corn stover by a microbial consortium. Although the manuscript presents an interesting topic, some major concerns deserve thorough attention before possible publication. The language of the article is kind of weak and needs a careful revision. Thus, authors are advised to improve the manuscript’s readability, merits, as well as its understanding for the journal readers.

Title
Contains repetitive phrases. It should be accurate and specific towards research aims.
Where are the keywords?

Introduction
The novelty of the work is missing from the introduction section of the manuscript. Further, the need for the study is also lacking in the current version. Authors can benefit from reading and citing the following articles: https://doi.org/10.1016/j.biombioe.2025.108054, https://doi.org/10.3390/biomass4030053, https://doi.org/10.1016/j.eti.2022.102459, https://doi.org/10.7717/peerj.11254.

There are several reports on the isolation of microbial strains from pig manure for lignocellulose degradation. How does your study extend the science in the relevant major?

Materials and methods
1. Sections 2.2 and 2.3: I don’t find any citation for relevant methods.
2. Lines 110-116: The preparation of standard sugars is a common procedure and can be deleted.
3. Line 186: Isn’t it the methodology section? Here, authors should rephrase the statement in terms of methodology while highlighting that maximum degradation was achieved after a certain period.
4. Where is the statistical analysis section?

Results
1. The numerical data is not given in all of the subsections of the heading 3.4.1. – 3.4.4.
2. Lines 253-255: Can you predict the species-level biodiversity of bacteria by targeting V3-V4 regions?
3. Authors should shed some light on the Relative abundance of the bacterial communities present in the consortium.
4. The legend of all figures needs to be elaborated.
5. Please explain the y-axis titles of the figure 3 components.
6. Figure 5 is not clear; it should be replaced with a better quality figure.
7. Line 291: Some data is not discussed in the discussion section. Also, the missing references (Dar et al, 2021) need to be added to the list.

Discussion
1. Lines 375-377: How does the author justify that the in vitro conditions used for experimentation are similar to the gut environment of the pig? Due to what factors does the abundance or the community structure of the microbial system remain unchanged?
2. From the whole discussion section, it is not clear how this study advances the scientific major and how its activities are superior to microbial consortia previously reported by other authors.

Conclusion
1. Precisely demonstrated the strengths and weaknesses of your study. And propose some future plans to tackle the weaknesses encountered in the present study.

Experimental design

-

Validity of the findings

-

Reviewer 3 ·

Basic reporting

-

Experimental design

-

Validity of the findings

-

Additional comments

General Comments: This study investigates the isolation of a strain with robust straw degradation capability from pig manure and its application in degrading corn stalks. The strain demonstrated promising degradation performance and production of key short-chain fatty acids. Furthermore, microbial composition and carbohydrate-active enzymes were characterized. However, there are still several issues in the current version of the manuscript that need to be addressed and improved. Major revisions are recommended before the manuscript can be considered for publication.

Specific Comments:

1. There are some grammatical errors and instances of awkward phrasing throughout the manuscript. It is strongly recommended that the authors carefully proofread the text and optimize the language.

2. The labels in the figures (e.g., “a” and “b” in Figure 1) are too small and unclear. Please revise to improve legibility.

3. The introduction of microbial treatment approaches is somewhat abrupt. Additionally, there are numerous successful examples of using anaerobic digestion reactors for agricultural waste treatment. The authors should better clarify the advantages and novelty of their study in comparison to these previous works.

4. The manuscript lacks proper control experiments, such as using a single-strain system or unacclimated microbial consortia. Without such comparisons, it is difficult to support the statement that "restricted culture is superior to single-strain" (Introduction, lines 78–80). It is recommended to include comparative data to substantiate this claim.

5. There are labeling errors in the manuscript, such as "4. Discussion" and "4. Conclusion." Please revise accordingly.

6. Supplementary materials (Figures S1–S4) are missing. These should be provided to complete the submission.

7. There is an inconsistency regarding the enzyme activity unit: Figure 1b uses "U/L," while Section 3.2 (line 219) refers to "U/mL." Please verify and ensure consistent use of the correct unit throughout the manuscript.

8. The reported lignin degradation rate is inconsistent: 56.83% is mentioned in the abstract, but 53.84% is stated in Section 3.3.6 (line 236, at 72 hours). Please verify the data and present it consistently.

9. The Discussion section does not sufficiently highlight the performance of the microbial community developed in this study. It lacks a thorough comparison with other agricultural waste degradation studies regarding degradation efficiency, key microbial taxa, and enzyme activities. Strengthening this aspect would greatly enhance the impact of the paper.

---

## Round 0.2 · Major Revisions

· Academic Editor

Major Revisions

I ask you to carefully correct the shortcomings pointed out by the reviewers.

·

Basic reporting

The research is able to provide the ability of a microbial system to break down lignocellulose in corn straw with SEM and FTIR analyzes. It has detected major microbial groups and common enzymes during degradation, and has obtained high rates of lignin, cellulose, and hemicellulose degradation, which opens up future possibilities of using it in making sustainable feed.

Experimental design

The research process was to collect microbial species in straw corn and then culture them to determine their capacity to break down lignocellulose. To give a complete picture of efficiency, structural changes and functional group alteration in the straw post degradation were analyzed using the scanning electron microscopy (SEM) and Fourier-transformation infrared spectroscopy (FTIR) measures.

Validity of the findings

Structural fractures and changes in functional groups are characterized by the consistent analysis SEM and FTIR and revealed to be damaged in corn straw that is degraded. The degradation rates are satisfactory with the past literature backing up its productiveness of the microbial system. Further, the discoveries on microbial phyla groups and enzymes ensured that the outcome is strong and can be used.

Additional comments

The novel is quite informative regarding microbial degradation of lignocellulose. Nevertheless, the inclusion of recent articles like the articles on https://doi.org/10.1002/appl.202300119 and https://doi.org/10.1016/B978-0-443-23679-2.00022-7 would make it more relevant.

Another area that can be considered to obtain further insight into degrading processes is studying higher pathways. Figures are presented well, and the manuscript can be accepted with some corrections.

Reviewer 3 ·

Basic reporting

no comment

Experimental design

no comment

Validity of the findings

no comment

Additional comments

The manuscript has been well revised according to the comments and is now acceptable for publication.

Reviewer 4 ·

Basic reporting

The paper presents valuable results on the improvement of corn straw utilization in order to diminish adverse environmental impacts of the straw burning under simultaneous benefits for animal husbandry. The manuscript is well-organized, written in good academic English. The ideas of authors are well described, comprehensible, and clear. References are relevant to the study.
In my opinion, Introduction is somewhat short, and it is advised to extend it on the authors' discretion.
The graphical materials are of a good quality, initial data are shared to get acquainted with the datasets used in the study.

Experimental design

The majority of the paragraph 2.9 describes different types of computational and bioinformatic analyses. Lines 204-215 describe bioinformatic analyses, which involve computational methods for analyzing biological data, particularly genetic and protein sequences. While bioinformatics often uses statistical principles, the specific processes described here – like using DIAMOND, KOBAS, KEGG, and dbCAN2 for sequence alignment and functional annotation – are not typically categorized as "statistical analysis". Pure statisctical analysis usually includes ANOVA tests, correlation, regression tests, etc. Therefore, a more accurate heading for the entire section might be "Data and Bioinformatic Analysis".
But looking at Fig.1-3, I see that each histogram has error bars, but there is no information in methodological section on how these error bars were calculated, and even what error do these error bars represent? Also, the letters. What do they represent? Do they outline the difference between the variables (e.g., ANOVA post-hoc test results) or something else? Please, add clear explanations in methodology.
Fig. 4 presents PCA results, but PCA procedure and algorithm are not described in methodology.
In general, the paper is interesting and scientifically sound, but methodological section related to computational and statistical procedures requires thorough revision and extension.

Validity of the findings

It is somewhat difficult to assess the validity of the findings until the authors provide complete information on methodological workflow. Discussion provides sufficient rumination on the study outcomes and provides relevant comparison with recent research work in this field. Conclusions are supported by the results.

---

## Round 0.3 · accepted · Accept

· Academic Editor

Accept

I ask you to remove the first two words "Investigating the" from the article title.

·

Basic reporting

The revised manuscript is to the point. Here by I am accepting the paper from my side.

Experimental design

To the point.

Validity of the findings

Now aligned.

Additional comments

Accepted.

·

Basic reporting

.

Experimental design

.

Validity of the findings

.

Additional comments

The authors have made substantial improvements to the manuscript and addressed all of the reviewer concerns. Hence, I recommend acceptance and publication of the manuscript in PeerJ.